# Exploring the Relationship between Lipid Profile, Inflammatory State and 25-OH Vitamin D Serum Levels in Hospitalized Patients

**DOI:** 10.3390/biomedicines12081686

**Published:** 2024-07-29

**Authors:** Sandica Bucurica, Andreea Simona Nancoff, Madalina Dutu, Mihaela Raluca Mititelu, Laura Elena Gaman, Florentina Ioniță-Radu, Mariana Jinga, Ionela Maniu, Florina Ruța

**Affiliations:** 1Department Gastroenterology, University of Medicine and Pharmacy Carol Davila, 020021 Bucharest, Romania; sandica.bucurica@umfcd.ro (S.B.); florentina.ionita-radu@umfcd.ro (F.I.-R.); mariana.jinga@umfcd.ro (M.J.); 2Department of Gastroenterology, University Emergency Central Military Hospital, 010825 Bucharest, Romania; andreea-simona.nancoff@rez.umfcd.ro; 3Department of Anesthesiology and Intensive Care, University of Medicine and Pharmacy Carol Davila, 020021 Bucharest, Romania; 4Department of Anesthesiology and Intensive Care, Dr. Carol Davila Central Military Emergency University Hospital, 010242 Bucharest, Romania; 5Department of Nuclear Medicine, University of Medicine and Pharmacy Carol Davila, 020021 Bucharest, Romania; 6Department of Nuclear Medicine, University Emergency Central Military Hospital, 010825 Bucharest, Romania; 7Department of Biochemistry, University of Medicine and Pharmacy Carol Davila, 020021 Bucharest, Romania; laura.gaman@umfcd.ro; 8Research Team, Pediatric Clinical Hospital Sibiu, 550166 Sibiu, Romania; ionela.maniu@ulbsibiu.ro; 9Department of Mathematics and Informatics, Faculty of Sciences, Lucian Blaga University Sibiu, 550012 Sibiu, Romania; 10Department of Community Nutrition and Food Safety, George Emil Palade University of Medicine, Pharmacy, Science and Technology of Targu Mures, 540142 Targu Mures, Romania; florina.ruta@umfst.ro

**Keywords:** dyslipidemia, vitamin D, inflammation: C-reactive protein, high-density lipoprotein cholesterol (HDL-c), cholesterol, triglycerides, low-density lipoprotein cholesterol (LDL-c)

## Abstract

Anomalies in lipid metabolism involve multifactorial pathogenesis, among other factors, being associated with an inflammatory state and disturbances in vitamin D status. The literature has focused on the binary relationships between inflammation and dyslipidemia, vitamin D and dyslipidemia, or vitamin D and inflammation. Our study aimed to explore the link between all these three factors: 25-OH vitamin D serum levels, the presence of inflammation assessed through serum C-reactive protein (CRP), and serum lipid profile in 2747 hospitalized patients. Our results showed a positive correlation of HDL-C with 25 (OH) vitamin D and a negative correlation of HDL-C with CRP. This relationship had different patterns in the statistical network analysis. The network analysis patterns are preserved for males and females, except for the relationship between CRP and vitamin D, which is present in male cases and absent in females. The same triangular relationship between all three—CRP, vitamin D, and HDL-C was found with different strengths of partial correlation in obese and non-obese patients. This pattern was similar in patients with and without fatty liver. A shifted pattern was found in the network analysis of hypertensive patients. The CRP was negatively correlated with vitamin D and HDL-C, and vitamin D was positively correlated with HDL-C in non-hypertensive patients. Castelli’s Risk indexes I and II were positively associated with CRP, suggesting that increased cardiovascular risk is proportional to an inflammatory state. The triad formed by altered serum lipid levels, inflammation, and vitamin D represents a complex relationship marked by specific dynamics between lipidic fractions such as HDL-C and C-reactive protein and vitamin D.

## 1. Introduction

Dyslipidemia, a metabolic condition marked by abnormal serum lipid levels, has seen a surge in prevalence over the past three decades. This condition is associated with an elevated risk of cardiovascular events, hepatic steatosis, or hypertriglyceridemia-induced pancreatitis [1,2,3,4]. Cholesterol (TC) and triglycerides (TG) are the two most clinically essential lipids that are found in plasma [5]. Two significant ratios are clinically used as predictors for cardiovascular diseases and atherogenicity—Castelli’s Risk Index I as the total cholesterol to high-density lipoprotein ratio (TC/HDL-C) and II which is the low-density lipoprotein to high-density lipoprotein ratio (LDL-C/HDL-C) [6]. Several studies concluded that these two indexes are easy and accessible markers that can evaluate the future risk of developing not only cardiovascular diseases but also they can correlate with the severity of coronary artery disease, diabetes, and athero-embolic stroke [7,8]. Dyslipidemia’s pathophysiology is complex, and it involves either genetic causes when discussing primary dyslipidemia, or it can be caused by a series of exogenous factors when considering secondary dyslipidemia. Therefore, diseases associated with changes in lipid metabolism should be seen as multifactorial [9] among the multiple factors involved in dyslipidemia, inflammation, and vitamin D influence lipid metabolism [10].

A chronic low-grade inflammatory state accompanies dyslipidemia [11]. Inflammation is usually associated with high levels of proinflammatory cytokines, such as TNF-α, TGF-1β, reactive species of oxygen (ROS), and interleukins (IL-6, IL-8), that have been proven to induce changes in lipid metabolism. These changes may refer to an increase in the production of liver lipids and downregulation of the clearance of lipid particles from the bloodstream [12,13]. Consequently, the proinflammatory interleukins induce the release of acute phase molecules, such as C-reactive protein (CRP) [12].

Regarding the association of inflammation and dyslipidemia, a recent study reported an inverse relationship between hs-CRP and an imbalanced serum lipid profile in patients with high cardiovascular risk [14]. Other data suggested that inflammation might partially mediate the association between dyslipidemia and cardiovascular diseases, showing a strong correlation between dyslipidemia and hs-CRP levels [15].

Vitamin D is a fat-soluble vitamin that has been shown to exert various roles, including anti-cancerous and anti-inflammatory effects. Vitamin D is linked to different chronic diseases such as diabetes types 1 and 2, metabolic syndrome, cardiovascular diseases, and autoimmune diseases, showing a pleiotropic effect [16,17,18]. Two large cohort studies on adolescent and pediatric populations from the United States of America and South Korea analyzed how vitamin D levels could influence cardiovascular risks. Vitamin D levels did not correlate with the serum lipid profile in both studies. Still, in their research, Reis JP et al. and Nam GE et al. reported that lower vitamin D levels were associated with specific cardiometabolic risks in adult life [19,20]. However, there are conflicting data regarding the dynamics between vitamin D and serum lipids. Some studies reported a significant positive association between serum cholesterol and vitamin D, while others found a weak or no association between total cholesterol and vitamin D [19,21,22]. In a large population study, females and males with lower levels of HDL-C had a higher risk of vitamin D deficiency (odd ratio males = 1.8 and odd ratio females = 1.4; both being > 1). This study reported no relevant association between LDL-C or TC with vitamin D deficiency [23]. In another research, Jiang et al. showed that higher vitamin D levels were associated with lower levels of HDL-C and LDL-C (*p* < 0.05) [24].

Regarding the relationship between inflammation and vitamin D, Dong Y et al. reported in their 4-year study that vitamin D supplementation was linked to lower levels of inflammation by decreasing the CRP concentration by 19%. This emphasizes that there is a delicate mechanism that keeps the balance between these lipid profiles, vitamin D, and inflammation levels, and external factors can potentially disrupt this balance [25].

The literature focused on the binary relationship between inflammation and dyslipidemia, vitamin D and dyslipidemia, or vitamin D and inflammation. The interplay between inflammation, vitamin D, and dyslipidemia is more complex as inflammation leads to changes in lipid metabolism, and vitamin D, through its immunomodulatory properties, can influence the inflammatory response. However, there is limited data available on this subject [26].

Our study aimed to explore the relationship between all three components by analyzing lipid profiles, C-reactive protein serum levels, and vitamin D status in hospitalized patients. 

## 2. Materials and Methods

This retrospective study included patients admitted between June 2020 and July 2022 at University Emergency Central Military Hospital Bucharest. Out of 3658 patients whose serum lipid profile, including total cholesterol, high-density cholesterol, triglycerides, C-reactive protein, and 25-OH vitamin D, were determined during admission, only 2747 patients met the inclusion criteria. 

The inclusion criteria were patient’s age > 18, admission as inpatients or outpatients, absence of renal failure, or serum creatinine < 1.2 mg/dL.

We excluded patients < 18 years old, pregnant patients, patients with sepsis, with renal failure (or serum creatinine > 1.2 mg/dL), and cases with missing data for serum total cholesterol, HDL-C, triglycerides, CRP, or 25-OH vitamin D. 

Demographic data, as well as data on obesity, fatty liver, hypertension, and diabetes mellitus, were recorded. (Figure 1)

The Sysmex XN3000 (Sysmex, Etten Leur, The Netherlands) was used for the hemogram. The Beckmann Coulter AU5822 analyzer (Beckman Coulter, Brea, CA, USA), using the spectrophotometry method, was used for the serum determination of albumin, total cholesterol, HDL cholesterol, triglycerides, aspartate aminotransferase (AST), alanine aminotransferase (ALT), and creatinine values, as well as immunoturbidimetry for CRP. For 25-OH vitamin D serum values, the chemiluminescence microparticle immunoassay (CMIA) method was used with the Abbott Alinity-i analyzer (Abbott Laboratories, IL, USA). The values for LDL cholesterol were automatically calculated using the Friedewald formula (LDL-C = TC − HDL-C − (TG/5) mg/dL) [27]. The upper normal limits, according to laboratory recommendations, were considered as follows: C-reactive protein 5 mg/L, 1.2 mg/dL for serum creatinine, and for 25-OH vitamin D, there were regarded as severely deficient below 10 ng/mL, deficient between 10 and 20 ng/mL, and sufficient > 20 ng/mL [28,29].

We also assessed Castelli’s Risk Index-I, which is defined as the TC/HDL-C ratio (<3.5 considered as low risk for cardiovascular diseases), and Castelli’s Risk Index II, which is defined as the LDL-C/HDL-C ratio (<3 regarded as low risk for cardiovascular diseases) [7,30].

The study was conducted according to the Declaration of Helsinki and approved by the Committee of Ethics from Central Emergency University Military Hospital No. 648/19/12/2023. Informed consent was obtained from all subjects involved for using and analyzing data for scientific purposes.

### Data Analysis

The data were reported as frequency (percentage) for qualitative variables and as median (IQR—interquartile range) or mean (SD—standard deviation) for quantitative variables. The normality of the data distribution was assessed using the Shapiro–Wilk test. Parametric and non-parametric tests were used to perform different group comparisons. Spearman correlation was used to determine the relationship between continuous variables. Network analysis, a multivariate analysis technique, was used to assess the relationship patterns between dyslipidemia, inflammatory variables, and vitamin D. The network nodes represent the laboratory variables. In contrast, the edges that connect the nodes represent the relationship between the variables (partial correlations between the variables, positive partial correlation—green color, and negative partial correlation—red color). Edge thickness indicates the degree of association between two nodes when the associations with all the other nodes were considered (thicker edges indicate stronger association). The data were analyzed using IBM SPSS^®^ (Statistical Package for the Social Science) v. 20 and R v.4.0.5 software (the R Foundation for Statistical Computing, Vienna, Austria, EGAnet package).

## 3. Results

### 3.1. Demographic and General Data

The cohort included 1849 females and 898 males (67.31% and 32.69%, respectively), with a mean age of 53.40 (SD = 16.16) years (Table 1). Regarding the metabolic status, the presence of one pathology associated with dyslipidemia was found in almost 25% of the individuals (24.43%), with fatty liver at a higher prevalence compared with obesity, diabetes mellitus, or hypertension (Table 1). The median value of CRP in our cohort was 2.69 mg/dL (IQR 1.11; 7.69), and 25 (OH) vitamin D had a median value of 27.10 ng/mL (IQR 19.20; 36.20). (Table 2). Median TC values were 190 mg/dL (IQR: 158.00–223.00), with a median Castelli’s Risk Index-I value of 3.6 (IQR: 2.97–4.41) and a Castelli’s Risk Index-II value of 2.15 (IQR: 1.62–2.80) (Table 1 and Table 2).

There was no significant difference related to vitamin D deficiency between fatty liver patients and non-fatty liver individuals (24.19% vs. 27.72%, *p* = 0.119). For the obese group, vitamin D deficiency was found in a slightly higher proportion (29.92%) than in the non-obese group (26.85%, *p* = 0.303). Overall, the patients with dysmetabolic status were found to be 26.82% with vitamin D deficiency compared with the individuals with no metabolic syndrome who presented serum 25-OH vitamin D levels below 20 ng/dL (27.21%) (*p* = 0.843) (Table 1).

### 3.2. Serum Lipid Profile and Status of Serum 25-OH Vitamin and Inflammation

When we analyzed the cut-offs for HDL-C, LDL-C, and total cholesterol, the C-reactive protein used as a marker of inflammation was higher in one-third of the cases (35.59%). Almost half of the patients (42.96%) had total cholesterol higher than 200 mg/dL, with 62.90% with LDL-C values above 100 mg/dL (Table 3). More than 80% of the subjects had normal liver enzyme profiles, with AST and ALT values below 35 UI/L. Most patients with total cholesterol lower than 200 had a CRP < 5 mg (63.43% vs. 36.67%, *p* = 0.000). In the group of subjects with high cholesterol levels, 69.83% (*n* = 824) had a non-inflammatory state, and 30.16% (*n* = 356) presented inflammation (*p* = 0.002) (Table 3).

The results showed a negative association between total cholesterol, the fractions (HDL-C and LDL-C), and the inflammatory status (CRP below and above 5 mg/L). The most relevant association was between CRP and HDL-C (r = −0.391, *p* = 0.000), followed by the atherogenic index, CRI-I, CRI-II, and triglycerides (Table 4, Figure 2).

Regarding the relationship with cholesterol fractions, the vitamin D level was found to be positively associated with total cholesterol, HDL, and LDL cholesterol fractions and negatively associated with serum triglycerides and Castelli’s Index Risk I and II, similar to the TG/HDL-C ratio (Table 5 and Table 6, Figure 1).

Patients with vitamin D deficiency had higher CRP levels (64.41% vs. 35.59%; *p* = 0.000) (Table 5).

The network nodes represent the laboratory variables, while the edges that connect the nodes represent the relationship between them (positive partial correlation—green color, negative partial correlation—red color). The thickness of the edges indicates the strength of the partial correlation between variables (TC_HDL-C_R—total cholesterol to high-density lipoprotein cholesterol ratio, LDLC_HDLC_R—low-density lipoprotein cholesterol to high-density lipoprotein cholesterol ratio, TG_HDLC_R—triglycerides to high-density lipoprotein cholesterol ratio, TC—total cholesterol, HDLC—high-density lipoprotein cholesterol, LDL-C—low-density lipoprotein cholesterol, CRP—C-reactive protein).

The network patterns are preserved for males and females, except for the relationship between CRP and vitamin D, which is present in male cases and absent in females. The same triangular relationship between all three—CRP, vitamin D, and HDL-C—was found with different strengths of partial correlation in obese and non-obese patients. This pattern was similar in patients with and without fatty liver (Figure 3).

A shifted pattern was found in the network analysis of hypertensive patients. The CRP was negatively correlated with vitamin D and HDL-C, and vitamin D was positively correlated with HDL-C in non-hypertensive patients. The vitamin D and CRP relationship changed into a positive correlation, while the other remained unchanged for the hypertension group. These results should be interpreted cautiously as there is a small number of study individuals, and this specific topic may represent the subject of future studies alone. 

## 4. Discussion

From a molecular perspective, new advances proved that specific signaling pathways were involved in the development of dyslipidemia [31,32]. All these molecular pathways have one thing in common—a significant impact on lipid metabolism. High levels of IL-1 and IL-6, found in chronic inflammatory or metabolic diseases, stimulate the synthesis of acute phase molecules in the liver, such as reactive C protein and fibrinogen [12,13].

In our study, the inflammatory state (characterized by CRP > 5 mg/L) was found in one-third of the patients with high total cholesterol. However, most individuals with total cholesterol below 200 mg/dL had lower CRP serum values. Although it was thought that dyslipidemia was linked to an increased risk of cardiovascular events, a recent cohort study conducted by Bafei et al. found that high CRP levels were a better predictor of cardiovascular diseases in patients with normal-range lipid profiles than those with lipid metabolism anomalies [14]. However, the literature contains controversial results regarding an explicit relationship between CRP and lipid profile. Data showed that serum lipid levels were unrelated to hs-CRP values, but there was a reported association between hs-CRP and the development of dyslipidemia [33]. Our study’s correlation between total cholesterol and CRP was negative (*p* = 0.000). Furthermore, when we analyzed the association of cholesterol lipoprotein fractions using the cut-offs for at-risk values with CRP, we found a weaker but significant correlation of CRP with LDL-C and a stronger negative association with HDL-C. For the group with TG > 150 mg/dL, most patients had no inflammation (*p* = 0.000), while almost 70% of the patients with CRP > 5 mg/dL had higher TG serum levels. The same pattern was found for LDL-C, with more than half of individuals with inflammation having high levels of LDL-C. However, this was not as relevant for values higher than 130 mg/dL when we found a similar proportion of inflammation versus non-inflammation (68.35% vs. 63.86%). We remarked that HDL-C levels were most relevant in their association with a non-inflammatory state, as almost all individuals in this study with protective levels of HDL-C also had a lower inflammatory status. However, a few subjects had normal range levels of HDL-C and high inflammatory status (26%). Similar results were reported by Vieira et al., where they found a possible link between vitamin D deficiency, inflammation, dyslipidemia, and cardiovascular risk among older people [34]. Conversely, HDL-C was the most relevant in its association with a non-inflammatory state, with 90.37% of individuals having protective levels of HDL-C when CRP < 5 mg/dL. Fewer patients had their HDL-C in a protective level when CRP exceeded 5 mg/dL (64.26%), with HDL-C showing a significant negative correlation with CRP (r = −0.391, *p* = 0.000).

In our study, Castelli’s Index risk factors I and II positively correlated with CRP. The TG/HDL-C ratio followed the same trend as Castelli’s Risk Index, with higher values as CRP increased (r = 0.386, *p* = 0.000, median 1.72 vs. 2.6). 

Vitamin D status showed a different relationship with dyslipidemia compared with CRP. Vitamin D deficiency was more prevalent in CRP > 5 mg/dL patients. We reported a negative correlation between vitamin D levels and serum levels of TG (r = −0.013) but a positive correlation with the lipid fractions—HDL-C, LDL-C, and total cholesterol levels. Similar results were reported in the literature [35,36]. In a study published by Elsheikh E. et al., vitamin D deficiency was found to be a possible cause of dyslipidemia. They showed that lower serum levels of 25(OH)D were strongly associated with changes in lipid metabolism, including TC, LDL-C, HDL-C, and TG [37]. Regarding the relationship between vitamin D and TC levels, similar results were found by Wang Y. et al., who reported a positive association between these two mentioned above and a negative association between LDL-C and vitamin D levels [38].

The multivariate network analysis showed a positive association between vitamin D and HDL-C, a negative association between CRP and HDL-C, and a negative association between CRP and vitamin D (Figure 2). The literature lacks data regarding the relationship between all these three parameters. One study reported that supplementary vitamin D did not influence triglycerides and HDL-C but did affect total cholesterol, LDL-C, and hs-CRP [10]. 

Regarding the network analysis for male and female subjects, we found a triad pattern between CRP, Vitamin D, and HDL-C in males. Still, the relationship between vitamin D and CRP disappeared in the network analysis for female cases. This could be explained by the influence of HDL-C structure on this lipoprotein’s function. In an article published by Wang X. et al., lipid kinetics differ between females and males, as HDL-C molecules have a larger size in females due to different rates of Apolipoprotein I and II synthesis compared to men [39]. Moreover, Chang C.-K. et al. further explored the association between serum CRP levels and HDL metabolites. Their study presented a new perspective on the inflammatory status and dyslipidemia by showing that chronic inflammation alters HDL-C structure and impairs its antiatherogenic–protective properties. Small and medium-sized HDL-C particles were strongly correlated with chronic inflammatory status [40]. This connection, which maintains a balance between the metabolic and inflammatory status in balance, is not yet fully understood. Therefore, we hope our study serves as a stepping stone for further studies. 

In the network analysis of this triad (CRP, vitamin D, and HDL-C) in patients with hypertension, we found a change from a negative (in non-hypertensive individuals) to a positive (in hypertensive individuals) correlation between CRP and vitamin D. This was also reported in a large study that included more than 17.000 subjects and showed an inverse association between vitamin D and CRP levels when 25(OH) vitamin D was deficient. This relationship became positive when vitamin D levels exceeded 21 ng/mL [41].

Our study’s strength lies in its large sample cohort and the complex associations uncovered between serum lipid fractions/risk indexes, CRP (as an inflammation marker), and serum 25-OH vitamin D through our network analysis. However, several limitations need consideration. Data on dyslipidemia treatments and vitamin D supplementation in these patients were unavailable. The population studied was heterogeneous, lacking a healthy control group, and we did not consider the causes that led to hospital admission and the occupation of the study participants. Additionally, the small sample size of participants with hypertension and diabetes limits the relevance of these subgroup results and warrants cautious interpretation.

Nevertheless, our findings demonstrated a positive correlation between serum high-density lipoprotein and 25-OH vitamin D serum levels, whereas an inverse correlation was observed between HDL-C and CRP. Furthermore, Castelli’s Risk indexes I and II were positively associated with CRP, indicating that increased cardiovascular risk correlates with an inflammatory state. 

## 5. Conclusions

The triad represented by altered serum lipid levels, inflammation, and vitamin D constitutes a complex relationship characterized by specific dynamics between lipid fractions (such as HDL-C), CRP, and vitamin D without discerning causality. Regarding clinical implications, our results showed that HDL-C is the lipid fraction primarily correlated with both 25-OH vitamin D and CRP. Furthermore, the fundamental physiological factors that regulate plasma lipid metabolism, including the differences between men and women, have yet to be clarified.

## Figures and Tables

**Figure 1 biomedicines-12-01686-f001:**
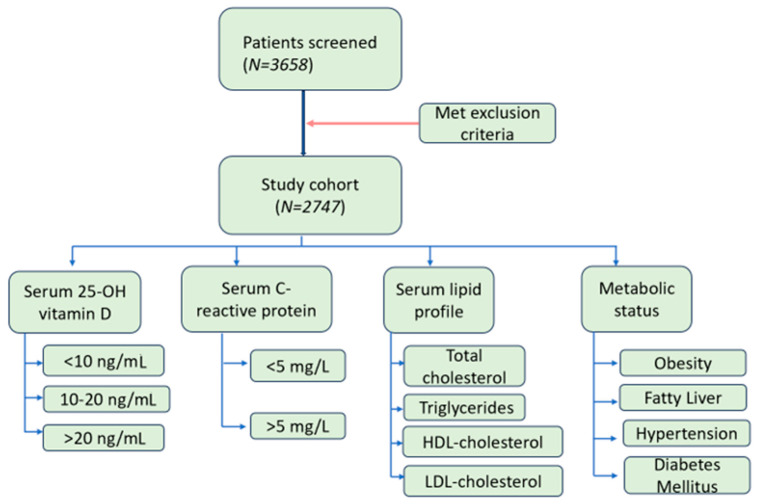
Flow diagram of the study cohort.

**Figure 2 biomedicines-12-01686-f002:**
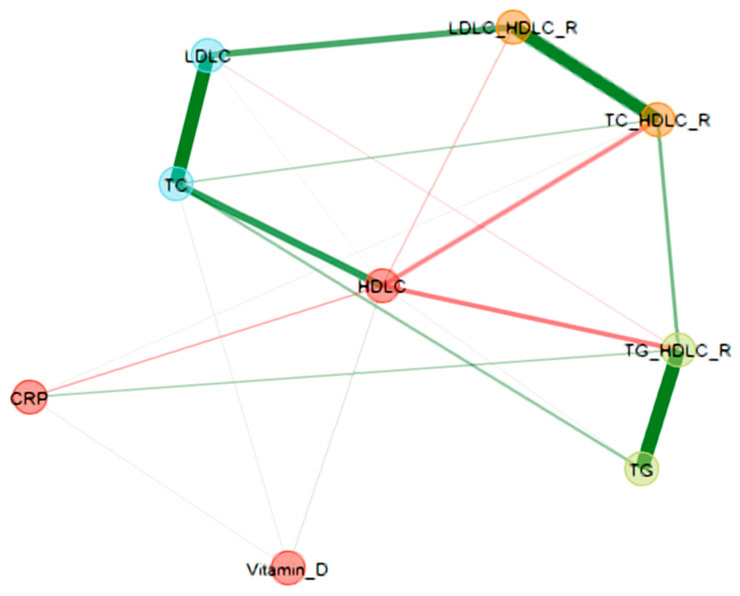
Network analysis of laboratory variables. The network nodes represent the laboratory variables, while the edges that connect the nodes represent the relationship between the variables (positive partial correlation—green color, negative partial correlation—red color). The thickness of the edges indicates the strength of the correlation. There is a positive partial correlation of HDL-C and TC with vitamin D. HDL-C was negatively correlated with CRP, TC/HDL-C (CRI I), TG/HDL-C, and LDLC/HDL-C (CRI II). A positive partial correlation was found between CRP and TC/HDL-C (CRI I), similar to that between CRP and TG/HDL-C. HDL-C and vitamin D presented a positive partial correlation, meanwhile CRP and vitamin D were negatively correlated.

**Figure 3 biomedicines-12-01686-f003:**
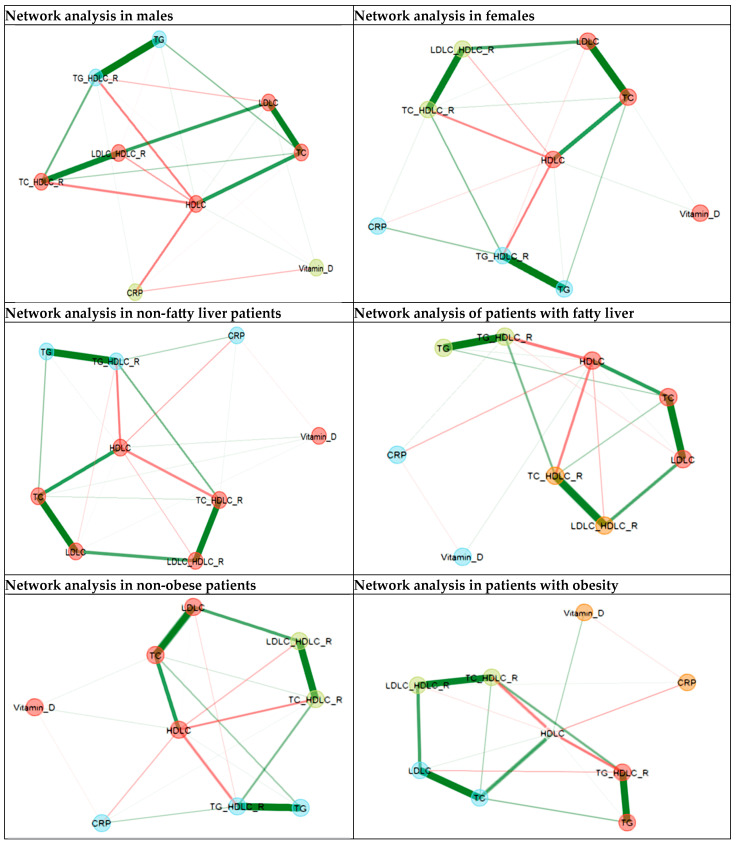
Network analysis by sex, obesity, hypertension, diabetes, and fatty liver. The network nodes represent the laboratory variables, while the edges that connect the nodes represent the relationship between the variables (positive partial correlation—green color, negative partial correlation—red color). Thickness of the edges indicates the strength of the correlation.

**Table 1 biomedicines-12-01686-t001:** Demographic data of analyzed population.

		All	Serum 25-OH Vitamin D (ng/dL)			
			≤10	10–20	>20	*p*-Value
Sex	Female	1849 (67.31)	72 (61.02)	414 (66.03)	1363 (68.08)	0.209 *
n (%)	Male	898 (32.69)	46 (38.98)	213 (33.97)	639 (31.92)	
Age (years) (M ± SD)		53.40 (16.16)	60.29 (17.13)	53.48 (17.78)	52.98 (15.47)	0.000 *
Fatty liver	No	2280 (83.00)	104 (88.14)	528 (84.21)	1648 (82.32)	0.172 *
n (%)	Yes	467 (17.00)	14 (11.86)	99 (15.79)	354 (17.68)	
Obesity	No	2503 (91.12)	108 (91.53)	564 (89.95)	1831 (91.46)	0.506 *
n (%)	Yes	244 (8.88)	10 (8.47)	63 (10.05)	171 (8.54)	
Hypertension	No	2735 (99.56)	116 (98.31)	624 (99.52)	1995 (99.65)	0.097 *
n (%)	Yes	12 (0.44)	2 (1.69)	3 (0.48)	7 (0.35)	
Diabetes mellitus	No	2723 (99.13)	116 (98.31)	624 (99.52)	1983 (99.05)	0.336 *
n (%)	Yes	24 (0.87)	2 (1.69)	3 (.48)	19 (.95)	
Metabolic syndrome	No	2076 (75.57)	92 (77.97)	473 (75.44)	1511 (75.47)	0.826 *
n (%)	Yes	671 (24.43)	26 (22.03)	154 (24.56)	491 (24.53)	

M ± SD—mean standard deviation, * *p* < 0.05 was considered statistically significant.

**Table 2 biomedicines-12-01686-t002:** Median values of analyzed serum parameters.

Serum Parameter	Median (IQR)
25(OH) vitamin D (ng/mL)	27.10 (19.20; 36.20)
C-reactive protein(CRP) mg/L	2.69 (1.11; 7.69)
Total cholesterol (TC) mg/dL	190.00 (158.00; 223.00)
High-density lipoprotein (HDL-C, mg/dL)	52.80 (43.30; 62.40)
Low-density lipoprotein (LDL-C, mg/dL)	113.00 (86.00; 141.00)
Serum Triglycerides (TG, mg/dL)	102.00 (74.00; 144.00)
Castelli Index I (CRI-I)	3.60 (2.97; 4.41)
Castelli Index II (CRI-II)	2.15 (1.62; 2.80)

**Table 3 biomedicines-12-01686-t003:** Lipid profile parameter cut-offs and inflammatory status results in the analyzed population.

		All	CRP (mg/L)		
			≤5	>5	*p*-Value
TC ≥ 200 (mg/dL)	No	1567 (57.04)	994 (54.68)	573 (61.68)	0.000 *
	Yes	1180 (42.96)	824 (45.32)	356 (38.32)	
ALB < 3.5 (g/dL)	No	2500 (91.01)	1796 (98.79)	704 (75.78)	0.000 *
	Yes	247 (8.99)	22 (1.21)	225 (24.22)	
HDL-C ≤ 40 (mg/dL)	No	2240 (81.54)	1643 (90.37)	597 (64.26)	0.000 *
	Yes	507 (18.46)	175 (9.63)	332 (35.74)	
LDL-C ≥ 100 (mg/dL)	No	1019 (37.10)	640 (35.20)	379 (40.80)	0.004 *
	Yes	1728 (62.90)	1178 (64.80)	550 (59.20)	
TG ≥ 150 (mg/dL)	No	2125 (77.36)	1476 (81.19)	649 (69.86)	0.000 *
	Yes	622 (22.64)	342 (18.81)	280 (30.14)	
ALT > 35 (U/L)	No	2229 (81.14)	1550 (85.26)	679 (73.09)	0.000 *
	Yes	518 (18.86)	268 (14.74)	250 (26.91)	
AST > 35 (U/L)	No	2317 (84.35)	1642 (90.32)	675 (72.66)	0.000 *
	Yes	430 (15.65)	176 (9.68)	254 (27.34)	
TC/HDL-C > 4.5	No	2114 (76.96)	1511 (83.11)	603 (64.91)	0.000 *
	Yes	633 (23.04)	307 (16.89)	326 (35.09)	

CRP—C-reactive protein, TC—total cholesterol, ALB—serum albumin, HDL-C—high-density lipoprotein cholesterol, LDL-C—low-density lipoprotein cholesterol, TG—serum triglycerides, ALT—alanine aminotransferase, AST—aspartate aminotransferase, TC/HDL-C ratio; * *p* < 0.05 was considered statistically significant.

**Table 4 biomedicines-12-01686-t004:** The association between lipid profile and serum C-reactive protein levels.

	r (*p*)	All	CRP (mg/L)		
			≤5	>5	*p*-Value
		Median (IQR)			
TC	−0.097	190.00	194.00	182.00	0.000 *
	(0.000)	(158.00; 223.00)	(165.00; 226.00)	(143.00; 217.00)	
TG	0.260	102.00	95.00	116.00	0.000 *
	(0.000)	(74.00; 144.00)	(70.00; 135.00)	(84.00; 162.00)	
HDL-C	−0.391	52.80	56.10	45.40	0.000 *
	(0.000)	(43.30; 62.40)	(47.60; 65.30)	(36.40; 55.00)	
LDL-C	−0.051	113.00	113.00	110.00	0.000 *
	(0.008)	(86.00; 141.00)	(89.00; 143.00)	(78.20; 138.00)	
CRI-I	0.298	3.60	3.44	3.98	0.000 *
	(0.000)	(2.97; 4.41)	(2.86; 4.15)	(3.26; 4.91)	
CRI-II	0.224	2.15	2.04	2.39	0.000 *
	(0.000)	(1.62; 2.80)	(1.56; 2.64)	(1.77; 3.10)	
TG/HDL-C	0.386	1.97	1.72	2.60	0.000 *
	(0.000)	(1.29; 3.11)	(1.16; 2.70)	(1.78; 3.96)	

TC—total cholesterol, TG—triglycerides, HDL-C—high-density lipoprotein cholesterol, LDL-C—low-density lipoprotein cholesterol, Castelli’s Risk Index-I-TC/HDL-C ratio, Castelli’s Risk Index II—LDL-C/HDL-C ratio, r—Spearman correlation coefficient, * *p* < 0.05 was considered statistically significant, IQR—interquartile range.

**Table 5 biomedicines-12-01686-t005:** Vitamin D status distribution and cut-off values of analyzed serum parameters.

		All	Serum 25-OH Vitamin D (ng/dL)			
			≤10	10–20	>20	*p*-Value
CRP > 5 (mg/L)	No	1818 (66.18)	42 (35.59)	378 (60.29)	1398 (69.83)	0.000 *
	Yes	929 (33.82)	76 (64.41)	249 (39.71)	604 (30.17)	
TC ≥ 200 (mg/dL)	No	1567 (57.04)	89 (75.42)	403 (64.27)	1075 (53.70)	0.000 *
	Yes	1180 (42.96)	29 (24.58)	224 (35.73)	927 (46.30)	
ALB < 3.5 (g/dL)	No	2500 (91.01)	71 (60.17)	532 (84.85)	1897 (94.76)	0.000 *
	Yes	247 (8.99)	47 (39.83)	95 (15.15)	105 (5.24)	
HDL-C ≤ 40 (mg/dL)	No	2240 (81.54)	63 (53.39)	467 (74.48)	1710 (85.41)	0.000 *
	Yes	507 (18.46)	55 (46.61)	160 (25.52)	292 (14.59)	
LDL-C ≥ 100 (mg/dL)	No	1019 (37.10)	63 (53.39)	268 (42.74)	688 (34.37)	0.000 *
	Yes	1728 (62.90)	55 (46.61)	359 (57.26)	1314 (65.63)	
TG ≥ 150 (mg/dL)	No	2125 (77.36)	82 (69.49)	491 (78.31)	1552 (77.52)	0.104 *
	Yes	622 (22.64)	36 (30.51)	136 (21.69)	450 (22.48)	
ALT > 35 (U/L)	No	2229 (81.14)	85 (72.03)	522 (83.25)	1622 (81.02)	0.016 *
	Yes	518 (18.86)	33 (27.97)	105 (16.75)	380 (18.98)	
AST > 35 (U/L)	No	2317 (84.35)	72 (61.02)	511 (81.50)	1734 (86.61)	0.000 *
	Yes	430 (15.65)	46 (38.98)	116 (18.50)	268 (13.39)	
TC/HDL-C > 4.5	No	2114 (76.96)	77 (65.25)	473 (75.44)	1564 (78.12)	0.003 *
	Yes	633 (23.04)	41 (34.75)	154 (24.56)	438 (21.88)	

CRP—C-reactive protein, TC—total cholesterol, ALB—serum albumin, HDL-C—high-density lipoprotein cholesterol, LDL-C—low-density lipoprotein cholesterol, TG—serum triglycerides, ALT—alanine aminotransferase, AST—aspartate aminotransferase, TC/HDL-C ratio, * *p* < 0.05 was considered statistically significant.

**Table 6 biomedicines-12-01686-t006:** The correlation between lipid profile and serum vitamin D status.

	r (*p*)	All	Serum 25-OH Vitamin D (ng/dL)			
			≤10	10–20	>20	*p*-Value
		Median (IQR)				
TC	0.149	190.00	162.50	181.00	195.00	0.000 *
	(0.000)	(158.00; 223.00)	(130.00; 199.00)	(147.00; 213.00)	(164.00; 227.00)	
TG	−0.013	102.00	101.50	97.00	102.00	0.446 *
	(0.498)	(74.00; 144.00)	(75.00; 161.00)	(72.00; 143.00)	(74.00; 144.00)	
HDL-C	0.181	52.80	41.55	50.50	53.90	0.000 *
	(0.000)	(43.30; 62.40)	(31.80; 54.30)	(39.90; 59.40)	(44.80; 63.40)	
LDL-C	0.128	113.00	85.90	107.00	115.00	0.000 *
	(0.000)	(86.00; 141.00)	(68.00; 119.00)	(80.00; 133.00)	(89.00; 144.00)	
CRI-I	−0.058	3.60	3.83	3.62	3.58	0.023 *
	(0.002)	(2.97; 4.41)	(2.98; 4.91)	(2.96; 4.49)	(2.97; 4.37)	
CRI-II	−0.020	2.15	2.17	2.14	2.15	0.673 *
	(0.295)	(1.62; 2.80)	(1.59; 3.01)	(1.61; 2.76)	(1.63; 2.78)	
TG/HDL-C	−0.100	1.97	2.51	2.00	1.92	0.000 *
	(0.000)	(1.29; 3.11)	(1.71; 4.82)	(1.32; 3.29)	(1.27; 3.02)	

TC—total cholesterol, TG—triglycerides, HDL-C—high-density lipoprotein cholesterol, LDL-C—low-density lipoprotein cholesterol, CRI-I—Castelli’s Risk Index–I, CRI-II—Castelli’s Risk Index–II, LDL-C/HDL-C ratio, r—Spearman correlation coefficient, *p*-value—statistical significance level, * *p* < 0.05 was considered statistically significant, IQR—interquartile range.

## Data Availability

Data availability statements are available upon reasonable request due to privacy restrictions.

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
