# Peer review of "Exploring the Relationship between Lipid Profile, Inflammatory State and 25-OH Vitamin D Serum Levels in Hospitalized Patients"

_biomedicines, 2024, doi:10.3390/biomedicines12081686_

Round 1

Reviewer 1 Report

Comments and Suggestions for Authors

This retrospective study (Manuscript ID #: Biomedicines-3056469) sought to explore the link between 25-OH vitamin D serum levels, the presence of inflammation (the level of serum C-reactive protein or CRP), and serum lepidic profile in 2747 patients with dyslipidemia. I have the following minor comments.

1.     The Introduction section detailedly describes the methods used in this study for measuring lipoprotein levels and Castelli’s Risk Index I/II. It is unnecessary because this study did not try to develop novel methods (the descriptions about the methods can be moved to the Materials and Methods section if the authors do not want to delete these descriptions). The Introduction section should focus on the previous studies about the relationships of vitamin D with inflammation and dyslipidemia, the current scientific question(s), and the authors’ hypothesis.

2.     Lines 60-62: LDL-C, HDL-C, IDL-C, VLDL-C, and Lp (a), instead of LDLC, HDLC, IDLC, VLDLC, and Lp a, are generally used in literature.

3.     The references should be inserted before, instead of after, a full stop at the end of a sentence.

4.     Lines 164-166: “Castelli’s Risk Index II, defined as the LDLC/HDLC Ratio” is repeated in this sentence.

Comments on the Quality of English Language

For example, "Our results showed that is a more complex ..." (line 29), "a thoroughly research" (line 35).

Author Response

Dear reviewer,

We want to thank you for taking the time to assess our manuscript and appreciate the effort you dedicated to providing feedback. We appreciate your comments, which will help improve our manuscript. We carefully considered your suggestions, addressed them, and highlighted them in the manuscript.

This retrospective study (Manuscript ID #: Biomedicines-3056469) sought to explore the link between 25-OH vitamin D serum levels, the presence of inflammation (the level of serum C-reactive protein or CRP), and serum lipidic profile in 2747 patients with dyslipidemia. I have the following minor comments.

  1. The Introduction section detailed the methods used in this study to measure lipoprotein levels and Castelli’s Risk Index I/II. It is unnecessary because this study did not try to develop novel methods (the descriptions about the methods can be moved to the Materials and Methods section if the authors do not want to delete these descriptions). The Introduction section should focus on the previous studies about the relationships of vitamin D with inflammation and dyslipidemia, the current scientific question(s), and the authors’ hypothesis.

Response: Dear reviewer, Thank you for suggesting reorganizing the introduction section. We have synthesized the content accordingly.

  1. Lines 60-62: LDL-C, HDL-C, IDL-C, VLDL-C, and Lp (a), instead of LDLC, HDLC, IDLC, VLDLC, and Lp a, are generally used in literature.

Response: We have performed the changes accordingly

  1. The references should be inserted before, instead of after, a full stop at the end of a sentence.

Response: We added the end of the sentences after inserting the references.

  1. Lines 164-166: “Castelli’s Risk Index II, defined as the LDLC/HDLC Ratio” is repeated in this sentence.

Response: We appreciate your keen eye for detail and have addressed the repetition in the content.

Sincerely yours,

The authors

Reviewer 2 Report

Comments and Suggestions for Authors

The submitted manuscript attempted to explore the relationship between inflammatory status and serum 25-OH vitamin D levels in patients with dyslipidemia. Although the study addresses an important and interesting topic, there are several significant concerns within the introduction, methodology, results, discussion, and overall presentation. This reviewer believes that this manuscript should benefit from substantial revisions. A major concern is the English language and scientific writing throughout this manuscript, which are not at an acceptable level. The writing contains grammatical errors and lacks clarity. The manuscript would benefit from thorough editing to improve the English language and scientific writing quality. Significant revisions are needed to enhance the clarity, coherence, and scientific rigor of the manuscript.

Abstract:

The abstract should be revised to align with the feedback provided for the other sections of the manuscript.

Introduction:

There is redundant information that does not align with the title and purpose of the study. For example, the introduction contains unnecessary information on lipid functions, lipoprotein classification, and the atherosclerosis process.

Additionally, the introduction fails to justify the need for the study or identify gaps in the current literature.

The rationale behind investigating the relationship between inflammatory status and serum 25-OH vitamin D levels in dyslipidemia is not clearly provided.

Materials and Methods:

Could you please provide a justification for not including a healthy control group in this study?

Additionally, cases with conditions such as fatty liver, hypertension, and diabetes mellitus were included in the statistical analysis of the collected data. Please include the methodological approach, particularly regarding statistical adjustments for potential confounding variables.

Results:

Please ensure that the results are presented in a manner that directly addresses the study's hypothesis and aligns with the title and purpose of this study.

How might factors such as age, gender, and other comorbidities influence the association between inflammatory status and serum 25-OH vitamin D levels in dyslipidemia?

According to this reviewer, the results do not adequately support the relationship between inflammatory status and serum 25-OH vitamin D levels in patients with dyslipidemia, as stated in the title of this study. Please clearly define the P-values (indicating the comparisons between variables) in all tables.

Discussion:

Some parts of the discussion are redundant, repeating detailed results that were already covered in the Results section.  For example, the correlation values and details of statistical significance could be summarized more succinctly to avoid redundancy.

The discussion should be more concise and focused.

Please ensure that the discussion closely aligns with the title, purpose, and relevant results of this study, providing a coherent narrative without redundant information.

Streamlining the discussion to focus more on interpreting the study’s findings, rather than reiterating results or unnecessary reviewing well-known pathways, would enhance readability and make the discussion interesting and relevant.

Additionally, please include concise information exploring the clinical implications of the findings.

Conclusion:

The conclusion is somewhat vague and lacks specific details from the study's results.

Additionally, it would be beneficial to briefly address implications for future research or clinical practice.

Comments on the Quality of English Language

Please see above (Comments and Suggestions for Author).

Author Response

Dear Reviewer,

We thank you for taking the time to assess our manuscript and appreciate the effort you dedicated to provide feedback. We appreciate your comments, which will help improve our manuscript. We carefully considered your suggestions, we did our best to address them and highlighted them in the manuscript.

The submitted manuscript attempted to explore the relationship between inflammatory status and serum 25-OH vitamin D levels in patients with dyslipidemia. Although the study addresses an important and interesting topic, there are several significant concerns within the introduction, methodology, results, discussion, and overall presentation. This reviewer believes that this manuscript should benefit from substantial revisions. A major concern is the English language and scientific writing throughout this manuscript, which are not at an acceptable level. The writing contains grammatical errors and lacks clarity. The manuscript would benefit from thorough editing to improve the English language and scientific writing quality. Significant revisions are needed to enhance the clarity, coherence, and scientific rigor of the manuscript.

Introduction:

There is redundant information that does not align with the title and purpose of the study. For example, the introduction contains unnecessary information on lipid functions, lipoprotein classification, and the atherosclerosis process. Additionally, the introduction fails to justify the need for the study or identify gaps in the current literature. – The rationale behind investigating the relationship between inflammatory status and serum 25-OH vitamin D levels in dyslipidemia is not clearly provided.

Response: We reorganized the introduction section. We have synthesized the content accordingly and focused on the relevant data from the literature.

Previously published studies focused on the relationship between:

  • inflammation and dyslipidemia,
  • vitamin D and dyslipidemia,
  • vitamin D and inflammation

In our study, we focused on highlighting the more complex interaction between inflammation, vitamin D status, and lipid profile (and this was the rationale for reminding these mechanisms).

Only a few studies have tried to illuminate this possible relationship, and as there are no studies in Eastern European countries, our study aimed to further explore this subtle interaction between these three components by analyzing a large population from Romania.

The study of R Jorde et al. (PMID: 20823896 DOI: 10.1038/ejcn.2010.176) “ High serum 25-hydroxyvitamin D concentrations are associated with a favorable serum lipid profile”- had no data about the inflammation status. Similarly, the study  “Association between serum vitamin D levels and lipid profiles: a cross-sectional analysis” (PMID: 38030665) published by Amir Gholamzad et al. studied only the relationship between vitamin D and lipid profile. The study of Baoshan Zhang (PMID: 38412162) also focused on these 2 vitamin D and  serum lipid – “The unique association between serum 25-hydroxyvitamin D concentrations and blood lipid profiles in agriculture, forestry, and fishing occupations: Insights from NHANES 2001-2014”

Materials and Methods:

Could you please justify not including a healthy control group in this study? 

Response:  Our study is a retrospective study that included hospitalized patients, and we had no access to a healthy control group. Also, there is no available data on vitamin D screening results in healthy populations in Romania. Moreover there are arguments against general healthy population screening for vitamin D . PMID 35406098, PMID 30721133, DOI 10.1002/jbm4.10417

Additionally, cases with conditions such as fatty liver, hypertension, and diabetes mellitus were included in the statistical analysis of the collected data. Please include the methodological approach, particularly regarding statistical adjustments for potential confounding variables.

Response:

We added separate analyses for males and females, as well as for hypertension, obesity, and diabetes. Network analysis, a multivariate analysis technique, was used to assess the relationship patterns between dyslipidemia, inflammatory variables, and vitamin D.

The network is formed by linking separate multiple regression models, taking into consideration partial correlations—correlations between two variables when controlling for all other variables—conditional association

 Partial correlations are closely related to coefficients from the multiple regression models: 

  • when an independent variable does not predict the dependent variable, we would not expect an edge in the network (the partial correlation is zero, no edge is drawn between two nodes, indicating that two variables are independent after controlling for all other variables in the network)
  • the edges connected to a single node show the expected result of a multiple regression analysis (a multiple regression analysis of a single dependent variable)
  • the network is formed by linking separate multiple regression models; partial correlation networks allow for mapping out the relationships among all variables (not only one dependent variable as in the case of a multiple regression analysis of a single dependent variable).

The formula for the partial correlation coefficient for X and Y, controlling for Z

Results:

Please ensure that the results are presented in a manner that directly addresses the study's hypothesis and aligns with the title and purpose of this study.How might factors such as age, gender, and other comorbidities influence the association between inflammatory status and serum 25-OH vitamin D levels in dyslipidemia?

Response: We added separate analyses for males and females and hypertension, obesity, and diabetes and reorganized the manuscript to align with its purpose.

According to this reviewer, the results do not adequately support the relationship between inflammatory status and serum 25-OH vitamin D levels in patients with dyslipidemia, as stated in the title of this study.

Response: We added details regarding the relationship between inflammatory status, 25 (OH) vitamin D, and lipid profile in hospitalized patients in the manuscript.

Disscution:

Some parts of the discussion are redundant, repeating detailed results already covered in the Results section.  For example, the correlation values and details of statistical significance could be summarized more succinctly to avoid redundancy.The discussion should be more concise and focused.

Please ensure that the discussion closely aligns with the title, purpose, and relevant results of this study, providing a coherent narrative without redundant information.Streamlining the discussion to focus more on interpreting the study’s findings, rather than reiterating results or unnecessary reviewing well-known pathways, would enhance readability and make the discussion interesting and relevant.Additionally, please include concise information exploring the clinical implications of the findings.

Response: Dear reviewer, We appreciate your time and effort in helping us improve our manuscript. Following your suggestions, we modified the section to be more clearly focused.

Conclusion:

The conclusion is somewhat vague and lacks specific details from the study's results. Additionally, it would be beneficial to briefly address implications for future research or clinical practice.

 Response:  We have revised the conclusion section and addressed implications for future research or clinical practice.

Reviewer 3 Report

Comments and Suggestions for Authors

Comments and Suggestions for Authors:

The manuscript by Bucurica and co-authors describes a retrospective study about the relationship between vitamin D levels, lipid profile and inflammatory markers in a large cohort of patients admitted to a hospital in Romania. The results of the study confirm the relationship between low vitamin levels, dyslipidemia and inflammatory state that is reported in the literature. In addition to the lack of originality, the study presents some points that in my opinion should be clarified. Language, presentation of data, appropriateness of references, and Figure 1 are acceptable. The interpretation of Figure 2 is problematic.

Some major points should be considered by the Authors

Any manuscript should start by acknowledging previous articles that covered the same topic and therefore should highlight the novel information or novel points that will be provided. Many articles have recently addressed the topic of the relationship between 25(OH)D and lipid profile, for example PMID 20823896, PMID 38030665, PMID 38412162, and others, and some of this literature perhaps deserves to be cited.

Page 2, lines 45-64 and page 3, lines 97-118. In the long introduction, a large part of the information provided consists in known concepts of the physiology of lipid metabolism or the pathogenesis of atherosclerosis. I think that in a scientific article this basic information should be conveniently summarized, thus shortening the introduction.

Page 3, lines 133-134. The investigation carried out in the study is much broader than what was stated in the last two lines of the introduction.

Page 3, lines 136-142. The population studied was very heterogeneous, and no information was provided regarding the causes that led to hospital admission. More importantly, the occupation of the study participants was not specified, although the relationship between vitamin D and lipid profile varies depending on the job (PMID 38412162). The possible intake of lipid-lowering drugs was also not specified.

Page 4, line 140. Diabetes mellitus is not one of the mandatory criteria for metabolic syndrome, since the presence of impaired glucose tolerance is sufficient. Furthermore, it would be better to specify which definition of metabolic syndrome has been adopted.

Page 6, Table 2. If I understood correctly, in participants with low 25(OH)D values, the total cholesterol level was also lower, which is at odds with most of the literature. How can the authors state that “vitamin D deficiency was found to be positively associated with total cholesterol and LDL cholesterol”? (page 5, lines 206-207).

Page 8, Figure 2. The interpretation of this figure is problematic. To begin with, I did not notice the negative correlation between CRP and TC which was evident from the results reported in Table 4. In addition, I did not notice the negative correlation (albeit not significant) between 25(OH)D and triglycerides reported in Table 2. Figure 2 is therefore inconsistent with the data  provided in the tables.

Page 9. The discussion leaves a lot to be desired. First, much of the discussion focuses on the relationship between inflammatory markers and lipid profile, completely losing sight of 25(OH)D. I do not deny that these results may be interesting, but they are off topic with the title of the manuscript. Second, in some previous studies no association was found between vitamin D and lipid profile, in both Western (PMID 19661053) and Eastern (PMID 23168294) countries, therefore the authors should explain why their results differ from these literature data. Third, the authors performed the analysis with the assumption that the relationship between the variables was linear (in fact they used correlation coefficients); however, it is known that the relationship between serum 25(OH)D and lipid profile often follows a nonlinear pattern (PMID 38412162) which requires a completely different statistical approach. Finally, given that some lipids such as HDL-C levels are strongly influenced by gender, a separate analysis in males and females would have been expected, otherwise this omission should be specified among the limitations of the study.

Minor remarks:

Language is not very accurate, and the manuscript is littered with inappropriate expressions such as "pathogeny" instead of "pathogenesis", in the abstract (line 26), “Upper normal limits for”, page 4, line 160; “incidence” instead of “frequency” or “prevalence”, page 5, line 193. I suggest having the text reviewed by a proofreading service.

Page 3, line 145. The acronyms AST, ALT must be moved to the first occurrence of the enzyme names i.e., to line 139, and not lines 155-156.

Page 5, line 185. I assume that R software was used for network analysis. In this case, specify the routine used.

Page 5, Table 1. Numbers less than 1.0 are somewhere reported with a leading zero (column 3) and elsewhere do not (columns 5-6). Please be consistent.

Page 7, line 223. “HLDC”. The authors probably intended “HDLC”.

Comments on the Quality of English Language

Moderate editing of English language required

Author Response

Dear reviewer,

We want to thank you for taking the time to assess our manuscript and appreciate the effort you dedicated to providing feedback. Your comments have been instrumental in improving our manuscript. We have carefully considered your suggestions, made the necessary revisions, and highlighted the changes in the manuscript; we did our best to address them.We are confident that these revisions will address your concerns and improve the quality of our manuscript.

Comments and Suggestions for Authors:

The manuscript by Bucurica and co-authors describes a retrospective study about the relationship between vitamin D levels, lipid profile and inflammatory markers in a large cohort of patients admitted to a hospital in Romania. The results of the study confirm the relationship between low vitamin levels, dyslipidemia and inflammatory state that is reported in the literature. In addition to the lack of originality, the study presents some points that in my opinion should be clarified. Language, presentation of data, appropriateness of references, and Figure 1 are acceptable. The interpretation of Figure 2 is problematic.

Some major points should be considered by the Authors

  1. Any manuscript should start by acknowledging previous articles that covered the same topic and therefore should highlight the novel information or novel points that will be provided. Many articles have recently addressed the topic of the relationship between 25(OH)D and lipid profile, for example PMID 20823896, PMID 38030665, PMID 38412162, and others, and some of this literature perhaps deserves to be cited. Page 2, lines 45-64 and page 3, lines 97-118. In the long introduction, a large part of the information provided consists in known concepts of the physiology of lipid metabolism or the pathogenesis of atherosclerosis. I think that in a scientific article this basic information should be conveniently summarized, thus shortening the introduction.

Response:

Previous published studies focused on the relationship between:

  • inflammation and dyslipidemia,
  • vitamin D and dyslipidemia,
  • vitamin D and inflammation

In our study, we focused on highlighting the more complex interaction between inflammation, vitamin D status, and lipid profile (and this was the rationale for reminding these mechanisms).

Also, we modified the content according to your suggestions by adapting it to be more concise.

The study of R Jorde et al. (PMID: 20823896 DOI: 10.1038/ejcn.2010.176), “ High serum 25-hydroxyvitamin D concentrations are associated with a favorable serum lipid profile,”- had no data about the inflammation status. Similarly, the study  “Association between serum vitamin D levels and lipid profiles: a cross-sectional analysis” (PMID: 38030665) published by Amir Gholamzad et al. studied only the relationship between vitamin D and lipid profile. The study of Baoshan Zhang (PMID: 38412162) also focused on these 2 vitamin D and  serum lipid – “The unique association between serum 25-hydroxyvitamin D concentrations and blood lipid profiles in agriculture, forestry, and fishing occupations: Insights from NHANES 2001-2014”

Only a few studies have tried to illuminate this possible relationship, and as there are no studies in Eastern European countries, our study aimed to further explore this subtle interaction between all these three components by analyzing a larger cohort.

We have included the references that you suggested.

  1. Page 3, lines 133-134. The investigation carried out in the study is much broader than what was stated in the last two lines of the introduction.

Response: We adjusted according to these suggestions.

  1. Page 3, lines 136-142. The population studied was very heterogeneous, and no information was provided regarding the causes that led to hospital admission. More importantly, the occupation of the study participants was not specified, although the relationship between vitamin D and lipid profile varies depending on the job (PMID 38412162). The possible intake of lipid-lowering drugs was also not specified.

Response: We added these as limitations of our study

  1. Page 4, line 140. Diabetes mellitus is not one of the mandatory criteria for metabolic syndrome since the presence of impaired glucose tolerance is sufficient. Furthermore, it would be better to specify which definition of metabolic syndrome has been adopted.

 Response:  We clarified this aspect in the manuscript

  1. Page 6, Table 2. If I understood correctly, in participants with low 25(OH)D values, the total cholesterol level was also lower, which is at odds with most of the literature. How can the authors state that “vitamin D deficiency was found to be positively associated with total cholesterol and LDL cholesterol”? (page 5, lines 206-207).

Response:

In  the PMID 20823896 article that you mentioned, there is a positive, significant association between TC and vitamin D

In the PMID 23168294 article that you mentioned, there is a positive tendency, not significant association (at p=0.05 level) between TC and vitamin D. 

We made the correction: “vitamin D was found to be positively associated with total cholesterol and LDL cholesterol.”

  1. Page 8, Figure 2. The interpretation of this figure is problematic. To begin with, I did not notice the negative correlation between CRP and TC which was evident from the results reported in Table 4. In addition, I did not notice the negative correlation (albeit not significant) between 25(OH)D and triglycerides reported in Table 2. Figure 2 is therefore inconsistent with the data provided in the tables.

Response:

We used a model of a network of partial correlations, not a model of a network of correlations. 

The edges that connect the nodes represent the relationship between the variables - partial correlations between the variables (not simple correlations).

Partial correlations are closely related to coefficients from the multiple regression models: 

  • when an independent variable does not predict the dependent variable, we would not expect an edge in the network (the partial correlation is zero, no edge is drawn between two nodes, indicating that two variables are independent after controlling for all other variables in the network)
  • the edges connected to a single node show the expected result of a multiple regression analysis (a multiple regression analysis of a single dependent variable)
  • the network is formed by linking separate multiple regression models; partial correlation networks allow for mapping out the relationships among all variables (not only one dependent variable as in the case of a multiple regression analysis of a single dependent variable).

The formula for the partial correlation coefficient for X and Y, controlling for Z

We decided to present the simple correlation values in a table so that we could compare them with those of other studies in the literature.

For the multivariate analysis, we used the network analysis technique to highlight (in a much more visually suggestive manner) the relationship between all variables: inflammation, vitamin D status, and lipid profile. A network is formed by linking separate multiple regression models, considering partial correlations – correlation between two variables when controlling for all other variables – conditional association

  1. Page 9. The discussion leaves a lot to be desired. First, much of the discussion focuses on the relationship between inflammatory markers and lipid profile, completely losing sight of 25(OH)D. I do not deny that these results may be interesting, but they are off topic with the title of the manuscript.

Response: We revised the discussion section

Second, in some previous studies, no association was found between vitamin D and lipid profile in both Western (PMID 19661053) and Eastern (PMID 23168294) countries. Therefore, the authors should explain why their results differ from these literature data.

Response: There are discrepancies between the results of different studies regarding the association between  vitamin D and lipid profile:

Moreover, conflicting data may result from different study designs and cohort characteristics. Also, the complex relationship between vitamin D and lipidic profile might be influenced by geographical region, UV exposure, lifestyle, diet, genetics, etc.

  1. Third, the authors performed the analysis with the assumption that the relationship between the variables was linear (in fact they used correlation coefficients); however, it is known that the relationship between serum 25(OH)D and lipid profile often follows a nonlinear pattern (PMID 38412162) which requires a completely different statistical approach.

Response: We used Spearman correlation (a correlation coefficient based on ranks) to assess the relationship between variables.

  1. Finally, given that some lipids, such as HDL-C levels, are strongly influenced by gender, a separate analysis in males and females would have been expected. Otherwise, this omission should be specified among the limitations of the study.

Response:  Following your valuable suggestion, we performed a separate analysis in males and females and added it to the manuscript.

Minor remarks:

Language is not very accurate, and the manuscript is littered with inappropriate expressions such as "pathogeny" instead of "pathogenesis", in the abstract (line 26), “Upper normal limits for”, page 4, line 

Response: We have corrected

160; “incidence” instead of “frequency” or “prevalence”, page 5, line 193. I suggest having the text reviewed by a proofreading service.

Response: We have corrected

Page 3, line 145. The acronyms AST, ALT must be moved to the first occurrence of the enzyme names i.e., to line 139, and not lines 155-156.

Response: We introduced these adequately

Page 5, line 185. I assume that R software was used for network analysis. In this case, specify the routine used.

R: We specified this in the manuscript

Page 5, Table 1. Numbers less than 1.0 are somewhere reported with a leading zero (column 3) and elsewhere do not (columns 5-6). Please be consistent.

R: We made the correction

Round 2

Reviewer 2 Report

Comments and Suggestions for Authors

This reviewer would like to thank the authors for their sincere efforts to improve the content and quality of the manuscript. However, the manuscript still requires significant revision for scientific writing. Many sentences are either improperly constructed or unclear, making it difficult to follow the scientific content. It is crucial to ensure that the manuscript is written in clear and concise English, with proper grammar and scientific terminology.

1. Let’s analyze the whole “Abstract” of this study, as an example:

First Two Sentences: The sentences “Dyslipidemia is a metabolic disorder with high burden worldwide. It is linked to increased cardiovascular risk and metabolic syndrome” are redundant in the context of an abstract. The purpose of the abstract is to provide a concise summary of the study, not to reiterate well-known facts.

The sentence “Lipidic metabolism anomalies involve a multi-perspective pathogenesis associated with an inflammatory state and disturbances in vitamin D status” is vague and does not provide a clear background or justification for the study.

The sentence on the aim of this study: The investigated variables should be presented in an order that aligns with the title and should remain consistent throughout the manuscript. The current order is inconsistent and needs revision.

The sentences on correlations: The sentence “Our results showed a positive correlation between serum high-density lipoprotein and 25-OH vitamin D serum levels, while an opposite correlation was found with CRP” lacks clarity regarding the correlation with CRP.

The statement “Moreover, Castelli’s risk index I and atherogenic index were positively associated with CRP, suggesting that increased risk is proportional to an inflammatory state” is not scientifically sound. It is unclear what the increased risk refers to.

The final sentence “This complex and multi-faceted relationship requires further research regarding causality” is not an appropriate conclusion for the study findings.

2. Introduction: The introduction has issues with scientific writing and coherence. The background should be presented more clearly and concisely, with a logical flow leading to the study objectives.

3. Materials and Methods: There are problems with English and scientific writing in this section, including repetition of information.

4. Results:

The results section is difficult to understand due to vague expressions and problems with scientific writing. For example, the statement “The vitamin D deficiency (< 20 ng/dL) was not significantly different in the metabolic groups, with an occurrence of relatively similar rates in the fatty liver patients compared with non-fatty liver individuals (4.19% vs. 27.72%, p=0.119)" is unclear.

The term “dysmetabolic status” may be not appropriate here.

Ensure that the results in the text correspond with the tables and figures. For instance, the mean age mentioned in the text does not match the table.

The titles of the tables are incomplete and should clearly indicate what the values in the table represent.

5. Discussion: The discussion section requires revision for English language and scientific writing. The arguments should be presented more coherently and logically.

6. Conclusions: The conclusions should be presented appropriately and in line with the study's findings.

7. Title: It might be more appropriate to use “lipid profile” instead of “lipidic profile” throughout the manuscript.

Comments on the Quality of English Language

Significant revisions are needed to address the issues with English and scientific writing.

Author Response

Dear Reviewer,

We are grateful for your time dedicated to reviewing our paper and helping us improve our manuscript. We appreciate your comments and have modified the content based on your insightful comments.

  1. Let’s analyze the whole “Abstract” of this study as an example:

First Two Sentences: The sentences “Dyslipidemia is a metabolic disorder with high burden worldwide. It is linked to increased cardiovascular risk and metabolic syndrome” are redundant in the context of an abstract. The purpose of the abstract is to provide a concise summary of the study, not to reiterate well-known facts. an inflammatory state and disturbances in vitamin D status” is vague and does not provide a clear background or justification for the study.

 Response: we have modified it to be more precise.

The sentence on the aim of this study: The investigated variables should be presented in an order that aligns with the title and should remain consistent throughout the manuscript. The current order is inconsistent and needs revision.

Response: We have performed this change

The sentences on correlations: The sentence “Our results showed a positive correlation between serum high-density lipoprotein and 25-OH vitamin D serum levels, while an opposite correlation was found with CRP” lacks clarity regarding the correlation with CRP.

 Response: We have corrected the sentence to regain its clarity

The statement “Moreover, Castelli’s risk index I and atherogenic index were positively associated with CRP, suggesting that increased risk is proportional to an inflammatory state” is not scientifically sound. It is unclear what the increased risk refers to.

 Response: These two are risk indexes related to cardiovascular risk; it is  included in their definition of the risk, but we changed the content

The final sentence “This complex and multi-faceted relationship requires further research regarding causality” is not an appropriate conclusion for the study findings.

Response: We have changed it accordingly.

  1. Introduction: The introduction has issues with scientific writing and coherence. The background should be presented more clearly and concisely, with a logical flow leading to the study objectives.

Response: We have reorganized the introduction as you well suggested

  1. Materials and Methods: There are problems with English and scientific writing in this section, including repetition of information.

Response: Thank you for your kind observation. We have modified the content

  1. Results:

The results section is difficult to understand due to vague expressions and problems with scientific writing. For example, the statement “The vitamin D deficiency (< 20 ng/dL) was not significantly different in the metabolic groups, with an occurrence of relatively similar rates in the fatty liver patients compared with non-fatty liver individuals (4.19% vs. 27.72%, p=0.119)" is unclear.

The term “dysmetabolic status” may be not appropriate here.

Ensure that the results in the text correspond with the tables and figures. For instance, the mean age mentioned in the text does not match the table.

Response: We made the suggested corrections

The titles of the tables are incomplete and should clearly indicate what the values in the table represent.

 Response: we have changed the titles

  1. Discussion: The discussion section requires revision for English language and scientific writing. The arguments should be presented more coherently and logically.

Response: We have structured the discussion section

  1. Conclusions: The conclusions should be presented appropriately and in line with the study's findings.

 Response: We introduced the data obtained in our research

  1. Title: It might be more appropriate to use “lipid profile” instead of “lipidic profile” throughout the manuscript.

Response: Thank you for your kind observation.  We have changed it according to your suggestion.

We have tried our best to respond to your comments and resubmitted an improved version of our manuscript.

Kind regards,

The authors

Reviewer 3 Report

Comments and Suggestions for Authors

Following my recommendations, the authors have significantly shortened the introduction, have clarified the statistical routines used, and have improved/integrated their network analysis of the variables, as well as corrected some few misspellings in the text. They also added, among the limitations, those regarding the heterogeneity of the sampled population. Overall, the revision carried out by the authors addressed my concerns and improved the quality of the manuscript. I have nothing else to ask.

Author Response

Dear Reviewer,

We are grateful for your support, and we appreciate the time dedicated for the revision of our paper.

With your help, we have an improved manuscript.

Kind regards,

The authors

Round 3

Reviewer 2 Report

Comments and Suggestions for Authors

This reviewer would like to thank the authors for their sincere efforts in further improving the quality of the manuscript. The reviewer has repeatedly requested that the authors to have the entire text proofread for English and scientific writing. It appears that the message was not fully conveyed, as the manuscript still requires improvement in English and scientific writing to reach an acceptable level. The improvements would enhance readability and clarify the findings.

It is not feasible for this reviewer to identify and explain every issue. However, as typical examples, the reviewer would like to provide the following three specific instances.

1.      L40-43: The authors wrote that “Moreover, Castelli’s risk index I and atherogenic index were positively associated with CRP, suggesting that an increased cardiovascular risk is proportional to an inflammatory state (CRI-I median of 3.44 when CRP<5 mg/dl vs. 3.98 median value for CRP>5 mg/dL (and r=0.298,p=0.000), CRI-II median of 2.04 when CRP<5 mg/dl vs. 2.39 median value for CRP>5 mg/dL (and r=0.224,p=0.000).” From the text used by the authors, it appears that Castelli’s risk index I and the atherogenic index are presented as different classes of indexes, which contrasts with lines 56-58 mentioned by the authors. Additionally, the text lacks consistent formatting for the indexes, statistical values and units. Moreover, there are redundant phrases that need to be removed. The above can be expressed as ‘Moreover, Castelli’s risk index (CRI)-I and -II were positively associated with CRP, suggesting that increased cardiovascular risk is proportional to an inflammatory state. The median CRI-I was 3.44 when CRP < 5 mg/dL compared to 3.98 when CRP > 5 mg/dL (r = 0.298, p < 0.001). Similarly, the median CRI-II was 2.04 when CRP < 5 mg/dL compared to 2.39 when CRP > 5 mg/dL (r = 0.224, p < 0.000).’

2.      L44-46: The text used by the authors may be expressed as ‘The triad of altered serum lipid levels, inflammation, and vitamin D represents a complex relationship characterized by specific dynamics among lipid fractions such as HDL-C, CRP, a marker of inflammation, and vitamin D, without establishing causality.’

3.     L122-127: The text used by the authors may be expressed as ‘Regarding the relationship between inflammation and vitamin D, Dong Y et al. reported in their 4-year study that vitamin D supplementation was associated with a 19% decrease in CRP concentration, indicating a relationship between vitamin D and reduced inflammation [25]. This emphasizes the intricate balance among lipid profiles, vitamin D, and inflammation levels, where external factors have the potential to disturb this delicate equilibrium.’

Some other important issues:

1.      As mentioned previously, the titles of the tables need to be improved by integrating necessary information. The title should properly reflect the content of the table and specify what the values in the table represent (for example, median and IQR). Please consult a relevant publication for guidance. By the way, what does ‘nr’ stand for in the title for Table 2?

2.      Please confirm the consistent use of terms throughout the entire manuscript, such as ‘lipid profile’ instead of ‘lipidic profile’.

Comments on the Quality of English Language

The authors should be asked to significantly improve the English and scientific writing throughout the entire manuscript.

Author Response

Dear reviewer,

We want to thank you for taking the time to reassess our manuscript and appreciate the continuous effort you dedicated to providing feedback. We carefully reconsidered your suggestions, addressed them, and highlighted them in the manuscript.

The reviewer has repeatedly requested that the authors to have the entire text proofread for English and scientific writing. It appears that the message was not fully conveyed, as the manuscript still requires improvement in English and scientific writing to reach an acceptable level.  -

We had English-proofed our manuscript by a native English speaker academic specialist.

  1. L40-43: The authors wrote that “Moreover, Castelli’s risk index I and atherogenic index were positively associated with CRP, suggesting that an increased cardiovascular risk is proportional to an inflammatory state (CRI-I median of 3.44 when CRP<5 mg/dl vs. 3.98 median value for CRP>5 mg/dL (and r=0.298,p=0.000), CRI-II median of 2.04 when CRP<5 mg/dl vs. 2.39 median value for CRP>5 mg/dL (and r=0.224,p=0.000).” From the text used by the authors, it appears that Castelli’s risk index I and the atherogenic index are presented as different classes of indexes, which contrasts with lines 56-58 mentioned by the authors. Additionally, the text lacks consistent formatting for the indexes, statistical values and units. Moreover, there are redundant phrases that need to be removed. The above can be expressed as–‘Moreover, Castelli’s risk index (CRI)-I and -II were positively associated with CRP, suggesting that increased cardiovascular risk is proportional to an inflammatory state. The median CRI-I was 3.44 when CRP < 5 mg/dL compared to 3.98 when CRP > 5 mg/dL (r = 0.298, p < 0.001). Similarly, the median CRI-II was 2.04 when CRP < 5 mg/dL compared to 2.39 when CRP > 5 mg/dL (r = 0.224, p < 0.000).’

Response: We would like to thank you for your keen eye, and we have made changes according to your kind suggestions.

  1. L44-46: The text used by the authors may be expressed as–‘The triad of altered serum lipid levels, inflammation, and vitamin D represents a complex relationship characterized by specific dynamics among lipid fractions such as HDL-C, CRP, a marker of inflammation, and vitamin D, without establishing causality.’

Response: We made changes according to your recommendations.

  1. L122-127: The text used by the authors may be expressed as–‘Regarding the relationship between inflammation and vitamin D, Dong Y et al. reported in their 4-year study that vitamin D supplementation was associated with a 19% decrease in CRP concentration, indicating a relationship between vitamin D and reduced inflammation [25]. This emphasizes the intricate balance among lipid profiles, vitamin D, and inflammation levels, where external factors have the potential to disturb this delicate equilibrium.’

Response: We made changes according to your recommendations.

Some other important issues:

  1. As mentioned previously, the titles of the tables need to be improved by integrating necessary information. The title should properly reflect the content of the table and specify what the values in the table represent (for example, median and IQR). Please consult a relevant publication for guidance. By the way, what does ‘nr’ stand for in the title for Table 2?

Response: We made changes according to your recommendations.

  1. Please confirm the consistent use of terms throughout the entire manuscript, such as ‘lipid profile’ instead of ‘lipidic profile’.

Response: We made changes according to your recommendations.

 Thank you once again for your patience and support in the process of publishing our research.

Sincerely,

The authors